# A Comparative Analysis of Toxicology and Non-Toxicology Care in Intoxicated Patients with Acute Kidney Injury

**DOI:** 10.3390/medicina60121997

**Published:** 2024-12-03

**Authors:** Chi-Syuan Pan, Chun-Hung Chen, Wei-Kung Chen, Han-Wei Mu, Kai-Wei Yang, Jiun-Hao Yu

**Affiliations:** 1Department of Emergency Medicine, China Medical University Hospital, Taichung 404, Taiwan; miss321@gmail.com (C.-S.P.); drumbeater1978@gmail.com (C.-H.C.); ercwk@mail.cmuh.org.tw (W.-K.C.); gackt0366@gmail.com (H.-W.M.); kaiwei.dennis.yang@gmail.com (K.-W.Y.); 2Department of Emergency Medicine, China Medical University Hsinchu Hospital, China Medical University, Hsinchu 302, Taiwan; 3Graduate Institute of Management, Chang Gung University, Taoyuan City 333, Taiwan; 4Department of Nursing, Yuanpei University of Medical Technology, Hsinchu 302, Taiwan

**Keywords:** intoxication, poisoning, acute kidney injury, toxicology, mortality, length of stay, nephrology

## Abstract

*Background and Objectives*: Intoxicated patients with acute kidney injury (AKI) experience high morbidity and mortality. While prior studies suggest that toxicology care settings improve outcomes, the impact of care settings on patients with AKI remains unclear. This study aimed to compare the outcomes of intoxicated patients with AKI managed in toxicology versus non-toxicology care settings. *Materials and Methods*: This retrospective cohort study included intoxicated patients admitted to a tertiary hospital between January 2022 and December 2023. Patients were categorized into toxicology and non-toxicology care settings. Demographic characteristics, clinical outcomes focusing on mortality and length of hospital stay, and evaluation scores were compared. *Results*: A total of 31 intoxicated patients with AKI were included in the study. There were no significant differences in mortality or hospital length of stay between toxicology and non-toxicology care settings. However, the mortality rate in the AKI group was significantly higher (16%) compared to intoxicated patients without AKI (1.9%). Additionally, hospital stays were consistently longer in the AKI group than in the non-AKI group across all age groups. *Conclusions*: Hospital length of stay and mortality did not differ significantly between toxicology and non-toxicology care settings for intoxicated patients with AKI. However, both hospital length of stay and mortality were notably higher in patients with AKI compared to those without AKI.

## 1. Introduction

Background: Intoxication remains a prevalent cause of emergency department (ED) visits, with acute kidney injury (AKI) posing a significant complication for affected patients. While emergency physicians typically determine whether hospitalization is necessary, toxicologists do not always manage patients with AKI; instead, nephrologists or general internists often provide care in non-toxicology wards or intensive care units (ICUs). We have inadequately explored how this variation in care impacts patient outcomes, especially in toxicology-specific settings. This study aims to evaluate the outcomes of intoxicated patients with AKI and assess the quality of care provided in toxicology wards, focusing on mortality rates and hospital length of stay for patients admitted to our institution.

AKI in intoxication are multifactorial and may be directly induced by toxins—such as medications, plants, animal venoms, chemicals, pesticides, and heavy metals—or may result secondarily from dehydration, shock, or other factors [1]. Accurate diagnosis of the specific intoxication is essential for determining appropriate treatment strategies for AKI. Therefore, identification is crucial [2]. Nephrologists are often consulted in ED and ICU to provide guidance on measures that enhance renal depuration of toxic agents [3]. In Taiwan, toxicologists may also be nephrologists or emergency physicians. At our hospital, the toxicologist has also been an emergency physician. Nephrologists are specialists in managing patients with renal failure and hemodialysis and may also have expertise in treating intoxicated cases [4], while toxicologists are experts in treating intoxicated cases. It raises the question of which ward provides better care for these patients.

Clinical research has observed that introducing a toxicology service for the ICU improves the quality of care [5]. However, researchers have not extensively explored the comparative analysis of care quality provided by toxicologists and nephrologists, underscoring the need for further investigation. According to studies, the number of intoxicated patients visiting the ED gradually decreased from 2006 to 2013; however, mortality rates, length of hospital stay, and medical costs did not show a corresponding decline [6]. Therefore, we need further research on how to provide appropriate care for patients with intoxicated. Collaborative care involving toxicology experts may suffice for managing these patients, and it is worth discussing whether specialized toxicology ward care is necessary. Some indicators and controversies exist in assessing the quality of medical care. We can use mortality rates and length of hospital stay as metrics to evaluate medical quality, as studies have demonstrated a relationship between these two factors [7]. Another study showed that shorter hospital stays are associated with better post-discharge outcomes, such as early rehospitalization and mortality [8]. Medical injuries during hospitalization lead to increased length of stay, costs, and mortality [5].

A study in Australia compared the average length of hospital stay and bed utilization before and after the establishment of a clinical toxicology unit, showing significant reductions. This saved over 2 million dollars [5]. Two other Australian studies showed that providing clinical toxicology services in the ED, wards, or ICUs can shorten hospital stays or days of ventilator use [5,9]. A study conducted in Hong Kong demonstrated that establishing an emergency toxicology team effectively reduced both the hospitalization rate and the length of hospital stay for acute intoxicated patients [10,11]. Additionally, another study revealed that a toxicology training unit in the ED led to a reduction in the hospital length of stay [11]. Both U.S. studies [12,13] indicated that the Medical Toxicology Inpatient Service and a toxicology admitting service reduced hospital days and costs. The latter study also showed that a medical toxicology admitting service could reduce mortality rates [13]. A systematic review of seven articles showed that the presence of inpatient toxicology services can reduce hospital stays and improve resource utilization [14].

In Taiwan, most hospitals lack specialized toxicology wards for managing intoxicated patients. Instead, nephrologists, who are experts in hemodialysis and skilled in metabolizing toxins, typically care for these patients, along with emergency physicians who excel in treating acute intoxicated cases. Intensive care physicians also manage severely poisoned patients in ICUs. It is important to note that internists and emergency physicians may approach thinking, diagnoses, and treatments differently.

Objective: This study aims to compare the quality of care provided to intoxicated patients with AKI in toxicology and non-toxicology settings. In toxicology settings, care is managed by toxicologists, while in non-toxicology settings, nephrologists are primarily responsible for patient management. The study evaluates whether outcomes, including mortality and length of hospital stay, differ between these settings. Additionally, outcomes for intoxicated patients with AKI are compared with those of intoxicated patients without AKI to provide broader context.

## 2. Materials and Methods

### 2.1. Study Design and Setting

This retrospective cohort study was conducted at China Medical University Hospital, a 2076-bed facility in Taiwan. The study included patients with intoxication admitted to ED between 1 January 2022, and 31 December 2023, who were subsequently admitted to the ward or ICU. Initial data were recorded by physician assistants and reviewed by toxicologists to ensure accuracy. Relevant data were extracted from the medical records of intoxicated patients, and follow-up continued until discharge, transfer, or death. The study was approved by the China Medical University Hospital Institutional Review Board under exemption code CMUH113-REC1-100 on 22 May 2024.

### 2.2. Study Population

Patients who presented to ED with suspected acute intoxication were initially included in the study. Patients admitted to a ward or ICU were required to consult a toxicologist or another specialist. The admission services were categorized into toxicology care settings (ward or ICU) and non-toxicology care settings (ward or ICU). Intoxication is defined as a pathological state resulting from the ingestion, inhalation, or exposure to a toxic substance that adversely affects normal physiological functions and requires medical evaluation or intervention [15]. AKI is defined as an increase in serum creatinine by 0.3 mg/dL or more within 48 h, an increase in serum creatinine to 1.5 times the baseline or more within the last 7 days, or urine output less than 0.5 mL/kg/h for 6 h [16].

Exclusion Criteria: We excluded patients if they were younger than 18 years, did not have AKI, were non-intoxication cases, were not admitted to the hospital, or were hospitalized in psychiatry for more than 5 days.

### 2.3. Study Outcomes

The primary outcomes evaluated were mortality rate, hospital length of stay (LOS), and ICU length of stay (ICU LOS).

### 2.4. Study Variables

The study considered various predictors and potential confounders, including age, gender, creatinine concentrations, estimated glomerular filtration rate and underlying diseases such as chronic kidney disease, diabetes mellitus, hypertension, mood disorders, coronary artery disease, cirrhosis, arrhythmia, cerebrovascular accident, and congestive heart failure. Additional factors analyzed included the Quick Sequential Organ Failure Assessment (qSOFA) score, Poisoning Severity Score (PSS) [17], recovery from acute kidney injury (AKI), the need for tracheal intubation, and hemodialysis.

### 2.5. Statistical Methods

We employed descriptive statistics to summarize the data, including means, medians, and standard deviations for continuous variables, as well as frequencies and percentages for categorical variables. Chi-square tests or Fisher’s exact tests were used for categorical variables, such as gender, underlying comorbidities, and mortality, while independent *t*-tests or Mann–Whitney U tests were applied to compare continuous variables, such as age, laboratory results, and length of stay, between two groups. Data were analyzed using SAS for Windows 9.4, and statistical significance was defined as *p* < 0.05.

## 3. Results

### 3.1. Patient Enrollment Flowchart

A total of 1343 patients presented to the ED with acute intoxication during the study period. We excluded 51 patients who were younger than 18 years, 37 non-intoxication cases, 539 patients without AKI, 15 patients remained hospitalized in the psychiatry department for over five days, and 670 patients who were not admitted to the hospital. Ultimately, 31 patients were enrolled in the study. Of these, 20 patients were treated in a toxicology setting (16 in the Toxicology ICU and 4 in the Toxicology Ward), while 11 patients were admitted to a non-toxicology setting (6 in the Non-Toxicology ICU and 5 in the Non-Toxicology Ward) (Figure 1).

### 3.2. Types of Substances Causing Intoxication in Toxicology and Non-Toxicology Settings

Table 1 shows that, among the 20 patients admitted to the toxicology ICU or ward, the substances included 4 cases of benzodiazepines, 2 of ethanol, 3 of antidepressants, 1 of valacyclovir, 3 of carbon monoxide, 1 of morphine, 2 of psychostimulants, 1 of pesticide, 1 of amlodipine, 1 of digoxin, 1 of baclofen, 1 of neutral detergent, and 1 of warfarin. In the non-toxicology setting, 11 intoxicated patients were admitted, including 2 cases of benzodiazepines and hypnotics, 2 of ethanol, 3 of antipsychotics, 2 of valacyclovir, 1 of morphine, 1 of pesticide, and 1 of metformin.

### 3.3. Demographic Characteristics

Table 2 presents the demographic characteristics, including age, gender, renal function, and underlying comorbidities, which did not show statistically significant differences between toxicology and non-toxicology care settings. Similarly, the recovery rate from AKI and treatment measures, such as tracheal intubation and emergent hemodialysis, also showed no statistical differences between the two groups. Additionally, neither the qSOFA score nor the poisoning severity score demonstrated significant differences.

### 3.4. Outcome

Table 3 compares the length of stay and mortality between toxicology and non-toxicology care settings. The mean ICU length of stay was 9.1 days for the toxicology group and 8.8 days for the non-toxicology group, with no significant difference (*p* = 0.951). Similarly, the mean hospital length of stay was 15 days for the toxicology group and 18 days for the non-toxicology group, also showing no significant difference (*p* = 0.679). For in-hospital mortality, there were 4 deaths (20%) in the toxicology group and 1 death (9.1%) in the non-toxicology group; however, this difference was not statistically significant (*p* = 0.62).

### 3.5. Comparative Analysis in Intoxicated Patients With or Without Acute Kidney Injury

Among intoxicated patients, those with AKI were compared to those without AKI. The average age was 63.3 years in patients with AKI compared to 47.2 years in those without (Table 4). The length of hospital stay was higher for patients with AKI (16.3 days) compared to those without AKI (5.5 days). Mortality was significantly higher in patients with AKI, with 5 out of 31 cases (16%) compared to 10 out of 539 cases (1.9%) in patients without AKI.

### 3.6. Subgroup Analysis of Length of Stay and Mortality Rates by Age Groups (≤40 Years, 41–64 Years, ≥65 Years)

In Table 5, we found that the length of hospital stay for patients with AKI was significantly longer than that for patients without AKI across different age groups. However, the mortality rate showed a significant difference only in the 41–64 age group.

## 4. Discussion

This study evaluated the outcomes of intoxicated patients with AKI managed in toxicology versus non-toxicology care settings. The analysis found no significant differences in hospital length of stay or mortality rates between the two care settings. However, intoxicated patients with AKI had significantly higher mortality rates and longer hospital stays compared to those without AKI. Notably, the prolonged hospital stays in the AKI group were consistent across all age groups, underscoring the substantial impact of AKI on the management and outcomes of intoxicated patients.

Our study demonstrated a significantly higher mortality rate of 16% in intoxicated patients with AKI compared to 1.9% in those without AKI. This finding aligns with evidence from a systematic review, which highlighted an association between AKI and increased mortality in intoxicated patients [18]. Among the five mortality cases in the AKI group, the substances involved included pesticides, organophosphates, neutral detergents, amlodipine, and digoxin. While these substances may not directly cause kidney injury, they can lead to renal dysfunction secondary to systemic effects, such as cardiovascular compromise, respiratory failure, or multiorgan dysfunction. The resulting decrease in renal function further impairs the elimination of toxic substances, creating a vicious cycle that exacerbates systemic toxicity, leads to multiorgan failure, and ultimately results in mortality [19,20,21].

While prior research has shown that toxicology inpatient services can improve hospital length of stay, cost, and mortality [5,12,13,14,22], our study did not demonstrate similar results when comparing toxicology and non-toxicology care settings. This discrepancy may be attributed to several factors. First, our study specifically focused on a subgroup of intoxicated patients with AKI, a condition associated with high mortality. The severe impact of AKI may have overshadowed or diminished any differences in outcomes between the two care settings, rendering them statistically insignificant. Second, the availability of critical care resources, such as hemodialysis or extracorporeal membrane oxygenation (ECMO), may vary between toxicology and non-toxicology ICU settings. Additionally, differences in the composition of multidisciplinary teams, including specialists and support staff, could influence outcomes. Third, the small sample size in our study may have limited the statistical power to detect significant differences. Despite these limitations, understanding the complex interactions between patient characteristics, care settings, and resource utilization is essential for optimizing the management of intoxicated patients. Further studies with larger cohorts are needed to explore whether specific care setting features provide distinct advantages.

The qSOFA score and PSS are commonly used tools for assessing the severity of poisoning, but their predictive accuracy varies. qSOFA, originally designed for sepsis [23], may not reliably predict outcomes in poisoning cases due to differences in pathophysiological mechanisms. Similarly, PSS provides a structured approach to categorizing poisoning severity, but its application can be inconsistent across diverse scenarios, which may affect its utility in clinical practice. An alternative tool, the New Poisoning Mortality Score (new-PMS), has been developed to predict mortality in acute poisoning The new-PMS incorporates demographic data, poisoning-related variables, and vital signs and has shown excellent predictive performance [24,25]. However, its complexity and reliance on multiple inputs may limit its practicality as a bedside tool in emergency settings, making qSOFA and PSS more feasible for routine use despite their limitations [26,27,28,29].

### Limitations

This study may be subject to selection bias because it includes patients from non-toxicology units, who have a higher prevalence of comorbidities. Nevertheless, we also admit toxicology patients with complications to the toxicology unit. The sample size is insufficient, which limits the generalizability and statistical power of our findings. The quality of care may vary between senior and junior residents, potentially affecting patient outcomes. Additionally, the quick Sequential Organ Failure Assessment (qSOFA) score may not accurately reflect the severity of poisoning cases.

## 5. Conclusions

There were no significant differences in hospital length of stay or mortality between toxicology and non-toxicology care settings for intoxicated patients with AKI. However, both hospital length of stay and mortality were higher in intoxicated patients with AKI compared to those without AKI.

## Figures and Tables

**Figure 1 medicina-60-01997-f001:**
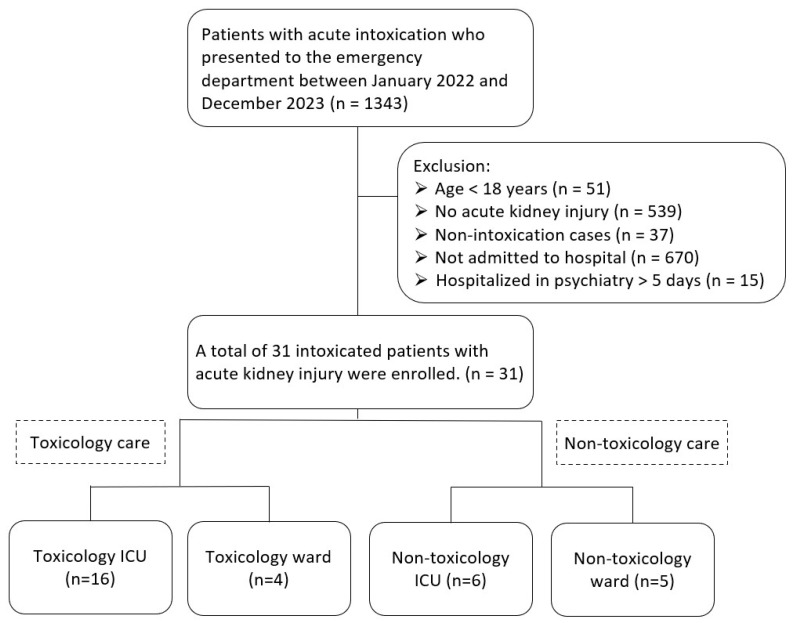
Patient enrollment flowchart.

**Table 1 medicina-60-01997-t001:** Types of substances causing intoxication in toxicology and non-toxicology settings.

Substances	Toxicology ICU/Ward (*n* = 20)	Non-Toxicology ICU/Ward (*n* = 11)	Total (*n* = 31)
Benzodiazepines and Hypnotics	4	2	6
Ethanol	2	2	4
Antidepressants	3	0	3
Antipsychotics	0	3	3
Valacyclovir	1	2	3
Carbon monoxide	3	0	3
Morphine	1	1	2
Amphetamines/psychoactive stimulants	2	0	2
Pesticide (Organophosphate)	1	1	2
Amlodipine	1	0	1
Digoxin	1	0	1
Baclofen	1	0	1
Neutral detergent	1	0	1
Warfarin	1	0	1
Metformin	0	1	1

Abbreviations: ICU—intensive care unit.

**Table 2 medicina-60-01997-t002:** Demographic characteristics in toxicology and non-toxicology care settings.

	Toxicology ICU/Ward (*n* = 20)	Non-Toxicology ICU/Ward (*n* = 11)	
	Mean ± SD/N (%)	Mean ± SD/N (%)	*p*
Age (years)	62.4 ± 21.7	64.8 ± 21.3	0.767
Male gender	13 (65.0)	5 (45.5)	0.288
Creatinine (mg/dL)	3.7 ± 2.2	4.3 ± 5.1	0.195
eGFR	21.6 ± 12	23.8 ± 15	0.398
Chronic kidney disease	5 (25)	4 (36.4)	0.109
Diabetes mellitus	5 (25)	3 (27.3)	0.727
Hypertension	11 (55)	7 (63.6)	0.455
Mood disorder	7 (35)	2 (18.2)	0.546
Coronary artery disease	1 (5)	2 (18.2)	0.586
Liver cirrhosis	1 (5)	1 (9.1)	0.909
Arrhythmia	2 (10)	1 (9.1)	0.586
Cerebrovascular accident	1 (5)	0 (0)	-
Congestive Heart Failure	0 (0)	1 (9.1)	-
Renal function recovery	12 (60)	8 (72.7)	0.356
Tracheal intubation	7 (35)	5 (45.5)	0.303
Emergent hemodialysis	8 (40)	7 (63.6)	0.167
qSOFA score	1.1 ± 0.7	0.6 ± 0.8	0.151
Poisoning severity score	3.1 ± 0.6	2.7 ± 0.7	0.092

Abbreviations: eGFR—estimated glomerular filtration rate; SD—standard deviation; ICU—intensive care unit; qSOFA—quick Sequential Organ Failure Assessment.

**Table 3 medicina-60-01997-t003:** Comparison of length of stay and mortality between toxicology and non-toxicology care.

	Toxicology ICU/Ward (*n* = 20)	Non-Toxicology ICU/Ward (*n* = 11)	
	Mean ± SD/N (%)	Mean ± SD/N (%)	*p*
Length of stay in ICU (day)	9.1 ± 10.5	8.8 ± 8.7	0.951
Length of stay in hospital (day)	15.0 ± 16.3	18.0 ± 24.3	0.679
In-hospital mortality	4 (20)	1 (9.1)	0.620

Abbreviations: ICU—intensive care unit; SD—standard deviation.

**Table 4 medicina-60-01997-t004:** Comparative Analysis in Intoxicated Patients With or Without Acute Kidney Injury.

	Intoxicated Patients with AKI (*n* = 31)	Intoxicated Patients Without AKI (*n* = 539)	
	Mean ± SD/N (%)	Mean ± SD/N (%)	*p*
Age (years)	63.3 ± 21.2	47.2 ± 20.0	<0.01 *
Length of stay in hospital (day)	16.3 ± 19.2	5.5 ± 8.9	<0.01 *
In-hospital mortality	5 (16)	10 (1.9)	<0.01 *

Abbreviations: SD—standard deviation; AKI—acute kidney injury; * A *p*-value of less than 0.05 was deemed statistically significant.

**Table 5 medicina-60-01997-t005:** Subgroup analysis of length of stay and mortality rates by age groups (≤40 years, 41–64 years, ≥65 years) comparing toxicology and non-toxicology care settings.

	Intoxicated Patients with AKI (*n* = 31)	Intoxicated Patients Without AKI (*n* = 539)	
	Mean ± SD/N (%)	Mean ± SD/N (%)	*p*
Age ≤ 40 (years)	30.3 ± 3.9	28.4 ± 6.4	0.225
Length of stay in hospital (day)	37.5 ± 36.5	5.3 ± 11.1	<0.01 *
In-hospital mortality	0/4 (0)	1/234 (0.4)	0.983
Age 41–64 (years)	52.1 ± 9.3	51.6 ± 6.9	0.469
Length of stay in hospital (day)	12.6 ± 16.5	6.9 ± 9.8	<0.05 *
In-hospital mortality	2/12 (16.7)	0/176 (0)	<0.01 *
Age ≥ 65 (years)	80.4 ± 11.8	75.6 ± 7.4	0.081
Length of stay in hospital (day)	13.6 ± 12	10.6 ± 16	<0.05 *
In-hospital mortality	3/15 (20)	9/129 (6.3)	0.076

Abbreviations: SD—standard deviation; AKI—acute kidney injury; * A *p*-value of less than 0.05 was deemed statistically significant.

## Data Availability

Data are available on reasonable request from the corresponding author. Restrictions may apply due to institutional policies, and ethical considerations.

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
