# Peer review of "A Comparative Analysis of Toxicology and Non-Toxicology Care in Intoxicated Patients with Acute Kidney Injury"

_medicina, 2024, doi:10.3390/medicina60121997_

Round 1
Reviewer 1 Report
Comments and Suggestions for Authors
Dear authors,
This paper evaluates treatment outcomes for patients with acute kidney injury (AKI) caused by toxins and provides several important findings. However, there are some areas for improvement and comments.
1. Patients from non-toxic units are included, and it is necessary to consider the impact of comorbidities. Clearly defining the study population and thoroughly evaluating patients' underlying conditions will enhance the interpretation of the results.
2. There is a lack of detail regarding specific treatment strategies and methods for both toxic and non-toxic treatments. Describing the differences in treatment protocols would clarify the comparisons.
3. It has been shown that the qSOFA score is unreliable for predicting mortality. Further research is needed to compare it with other scores and explore more appropriate indicators.
4. A deeper discussion on the impact of polypharmacy on treatment outcomes would be beneficial. Detailed data on the types of concomitant medications and their interactions should be included.
5. While the high mortality rate in AKI patients is significant, discussing the factors and context in comparison to other studies could provide further insights.
6. It is also necessary to address the impact of age differences on the outcomes. Comparing results across different age groups and providing a more detailed analysis would be beneficial.
7. Are you considering evaluating other scoring systems? Do you plan to compare your results with other mortality prediction scores, such as SOFA or PSS?
Author Response
Dear Reviewer:Thank you for your valuable comments and suggestions regarding our manuscript. We appreciate the time and effort put into reviewing our work and are grateful for your insightful feedback, which has helped us enhance the clarity and robustness of our study. Below, we provide a point-by-point response to address each concern raised.
1: Patients from non-toxic units are included, and it is necessary to consider the impact of comorbidities. Clearly defining the study population and thoroughly evaluating patients' underlying conditions will enhance the interpretation of the results.
We appreciate your suggestion and have revised the manuscript (methods inclusion criteria) accordingly. We now clearly define our study population as poisoning patients with acute kidney injury (AKI). For a standardized AKI definition, we use criteria from the Acute Kidney Injury Network (AKIN): (1) a serum creatinine increase of 0.3 mg/dL or more within 48 hours, (2) a serum creatinine increase to 1.5 times baseline or higher within seven days, or (3) urine output less than 0.5 mL/kg/h for six hours. (Reference 15: Mehta et al., 2007).
Additionally, we have carefully reviewed the patients' underlying conditions and included these comorbidities in Table II to give readers a more comprehensive understanding of the health background of each patient.
- There is a lack of detail regarding specific treatment strategies and methods for both toxic and non-toxic treatments. Describing the differences in treatment protocols would clarify the comparisons.
- It has been shown that the qSOFA score is unreliable for predicting mortality. Further research is needed to compare it with other scores and explore more appropriate indicators.
- A deeper discussion on the impact of polypharmacy on treatment outcomes would be beneficial. Detailed data on the types of concomitant medications and their interactions should be included.
- While the high mortality rate in AKI patients is significant, discussing the factors and context in comparison to other studies could provide further insights.
AKI in poisoning cases may not only reflect renal failure but also the impact of other organ dysfunctions on the kidneys. In our deceased patients, we observed complications such as respiratory failure, heart failure, septic shock, and liver failure. Generally, in poisoning patients without renal failure, the toxin is primarily metabolized by the kidneys. However, renal failure can exacerbate the toxic effects of the substance. AKI is commonly observed in critically ill patients and may result from secondary events. Severe illness can lead to AKI, which in turn may contribute to increased mortality.
- It is also necessary to address the impact of age differences on the outcomes. Comparing results across different age groups and providing a more detailed analysis would be beneficial.
- Are you considering evaluating other scoring systems? Do you plan to compare your results with other mortality prediction scores, such as SOFA or PSS?
Thank you once again for your constructive comments.
Sincerely,
Jiun-Hao Yu, MD
Attending Physician and Director of Emergency Medicine
Department of Emergency Medicine, China Medical University Hsinchu Hospital, China Medical University, Hsinchu 30272, Taiwan
E-mail address: flykingyu@gmail.com
Reviewer 2 Report
Comments and Suggestions for Authors
Dear authors, after reading your paper, I think there are several drawbacks
1. I think you should define "poisoning" better. A medication overdose is poisoning?
2. Is alcohol linked to AKI? Please give more data. Are you referring to ethanol or methanol?
3. You have missed the essential in your analysis. What was the serum creatinine and/or eGFR at admission? Did they recover the renal function? How many needed hemodialysis? Did the admission in a specific ward influenced these parameters? Considering that you have a small sample, you should provide supplementary data.
Author Response
Dear Reviewer:Thank you for your valuable comments and suggestions regarding our manuscript. We appreciate the time and effort put into reviewing our work and are grateful for your insightful feedback, which has helped us enhance the clarity and robustness of our study. Below, we provide a point-by-point response to address each concern raised.
- I think you should define "poisoning" better. A medication overdose is poisoning?
- Is alcohol linked to AKI? Please give more data. Are you referring to ethanol or methanol?
References: Alcohol Res. 2017;38(2):283–288. Alcohol Misuse and Kidney Injury: Epidemiological Evidence and Potential Mechanisms.
At-Risk Drinking Is Independently Associated With Acute Kidney Injury in Critically Ill Patients.
- You have missed the essential in your analysis. What was the serum creatinine and/or eGFR at admission? Did they recover the renal function? How many needed hemodialysis?
We will add serum creatinine, eGFR, need for hemodialysis, and renal recovery data in the Results section, specifically in Table 2. We will also analyze how admission to a specific ward influences these parameters. Additionally, we will provide supplementary data.
Thank you once again for your constructive comments.
Sincerely,
Jiun-Hao Yu, MD
Attending Physician and Director of Emergency Medicine
Department of Emergency Medicine, China Medical University Hsinchu Hospital, China Medical University, Hsinchu 30272, Taiwan
E-mail address: flykingyu@gmail.com
Round 2
Reviewer 2 Report
Comments and Suggestions for Authors
Dear authors, I have noticed the extensive modification you have performed. The only issue that I still do not completely understand is ethanol induced AKI. By my knowledge, is condition is extremely rare, taking into account that ethanol is one of the most consumed drinks in the world. Meanwhile, you did not register any case of methanol or ethilenglicol intoxication; which are more likely to lead to AKI